# Synergistic Pro-Apoptotic Effect of a Cyclic RGD Peptide-Conjugated Magnetic Mesoporous Therapeutic Nanosystem on Hepatocellular Carcinoma HepG2 Cells

**DOI:** 10.3390/pharmaceutics15010276

**Published:** 2023-01-13

**Authors:** Xuanping Zhao, Chuan Liu, Zichao Wang, Yingyuan Zhao, Xuyang Chen, Haizhen Tao, Hong Chen, Xueqin Wang, Shaofeng Duan

**Affiliations:** 1College of Bioengineering, Henan University of Technology, Zhengzhou 450001, China; 2Key Laboratory of Functional Molecules for Biomedical Research, Henan University of Technology, Zhengzhou 450001, China; 3Institute for Innovative Drug Design and Evaluation, School of Pharmacy, Henan University, Kaifeng 475004, China

**Keywords:** cyclic RGD peptide, anticancer drug delivery, targeted nanoparticles, hepatocellular carcinoma, apoptosis

## Abstract

Numerous nanocarriers have been developed to deliver drugs for the treatment of hepatocellular carcinoma. However, the lack of specific targeting ability, the low administration efficiency, and insufficient absorption by hepatocellular carcinoma cells, severely limits the therapeutic effect of the current drugs. Therefore, it is still of great clinical significance to develop highly efficient therapies with few side effects for the treatment of hepatocellular carcinoma. Herein, we developed a highly effective nanocarrier, cyclic RGD peptide-conjugated magnetic mesoporous nanoparticles (^RGD^SPIO@MSN NPs), to deliver the chemotherapeutic drug doxorubicin (DOX) to human hepatocellular carcinoma HepG2 cells, and further explored their synergistic apoptosis-promoting effects. The results showed that the prepared ^RGD^SPIO@MSN NPs had good stability, biosafety and drug-loading capacity, and significantly improved the absorption of DOX by HepG2 cells, and that the ^RGD^SPIO@MSN@DOX NPs could synergistically promote the apoptosis of HepG2 cells. Thus, this cyclic RGD peptide-modified magnetic mesoporous silicon therapeutic nanosystem can be regarded as a potentially effective strategy for the targeted treatment of hepatocellular carcinoma.

## 1. Introduction

According to the Global Cancer Statistics Report 2022, hepatocellular carcinoma (HCC) is one of the most common tumors and the fourth leading cause of malignancy-related deaths [1]. Currently, the primary therapeutic methods for HCC are still surgery and chemotherapy [2,3]. Although the incidence of liver cancer has remained stable after decades of slow decline, the survival rate of patients is still low because of the high recurrence and metastasis rate, poor prognosis, the systemic toxicity and side effects of chemotherapy drugs, and the lack of satisfactory diagnosis and treatment strategies with high targeting ability [4,5,6,7,8,9]. Therefore, it is indispensable to develop new nanocarriers with good biosafety, high drug-loading capacity, and robust targeting action against liver cancer [10,11,12].

Doxorubicin (DOX) is a broad-spectrum antitumor drug with good antitumor activity in the clinical treatment of HCC [13,14,15,16]. However, due to the lack of targeting ability and potential cytotoxicity, HCC patients always suffer from different degrees of side effects after clinical medication, especially cardiotoxicity, thereby limiting its clinical application [17,18,19,20]. Therefore, it is essential to search for an ideal targeted drug delivery system for DOX.

Superparamagnetic iron oxide nanoparticles (SPIO NPs) (mainly Fe_3_O_4_ and γ-Fe_2_O_3_ NPs), with diameters ranging from 1 to 100 nm, have been employed as anticancer drug delivery vehicles in targeted tumor therapies due to their nanometer size effect, inherent superparamagnetism, and amenability to functionalized modification [21,22,23]. However, bare SPIO NPs without a surface envelope or decoration tend to aggregate and precipitate in solution, which severely impedes their application in vitro and in vivo. In this regard, bare SPIO NPs should be decorated to enhance their stability and biocompatibility, for further use as a carrier of chemotherapy drugs in antitumor therapy [24,25].

Mesoporous silica, a highly porous material, has attracted extensive interest as a coating layer for SPIO NPs due to its low cytotoxicity, stability, excellent biodegradability and biocompatibility, high drug-loading and delivery performance, and controlled drug release [26,27,28,29]. Herein, mesoporous silica was employed to achieve favorable dispersibility, stability, and biocompatibility of SPIO NPs. We hypothesize that, as drug nanocarriers, mesoporous-silica-decorated SPIO NPs can remarkably improve drug delivery efficiency, increase the drug concentration in tumor cells or tumor tissues, reduce side effects, and overcome the disadvantages of traditional antitumor drugs in cancer treatment.

RGD peptide (arginine-glycine-aspartate, Arg-Gly-Asp) is an integrin α_v_β_3_-targeting ligand and integrin receptors are known to be highly expressed in most tumor cells [30,31,32,33,34]. Therefore, RGD peptide is a promising targeted capping ligand for drug delivery in cancer therapy. In addition, compared with endogenous linear RGD peptides, which have a very short half-life, cyclic RGD peptides containing two disulfide bonds have a more stable structure and a stronger affinity [35,36,37]. Hence, cyclic RGD peptide-conjugated nanocarriers can be employed as an effective targeted drug delivery system for the synergistic treatment of HCC.

Herein, we constructed a targeted nanocarrier based on cyclic RGD peptide-conjugated magnetic mesoporous nanoparticles, for the effective delivery of the antitumor drug DOX to treat hepatocellular carcinoma HepG2 cells, and further explored its synergistic pro-apoptotic effects. Specifically, the SPIO NPs served as the core, and SPIO@MSN NPs were fabricated by modification of the surface with mesoporous silica. Subsequently, the SPIO@MSN NP carriers were loaded with the antitumor drug DOX and then conjugated to cyclic RGD peptide to construct DOX-loaded MSN@DOX NPs (^RGD^SPIO@MSN@DOX NPs), which were utilized to promote apoptosis of human hepatocellular carcinoma HepG2 cells. The performance of the fabricated ^RGD^SPIO@MSN@DOX NPs was evaluated in terms of morphology, particle size, stability in different media, biocompatibility, and drug loading, etc. In addition, the effects of ^RGD^SPIO@MSN@DOX NPs on hepatocellular carcinoma HepG2 cells, including anti-proliferative ability, cell viability, and apoptosis rate, were also investigated (Figure 1). The results showed that ^RGD^SPIO@MSN@DOX NPs clearly improved cellular uptake of DOX, inhibited cell growth, and significantly induced apoptosis in the treated HepG2 cells, indicating that the prepared ^RGD^SPIO@MSN@DOX NPs are a potential antitumor drug delivery vector for inhibiting the growth of hepatocellular carcinoma HepG2 cells.

## 2. Materials and Methods

### 2.1. Materials

Cyclic RGD peptide was obtained from Shanghai Jill Biochemical Co., Ltd. (Shanghai, China). DOX, Hexadecyl trimethyl ammonium Bromide (CTAB), Tetraethyl orthosilicate (TEOS), and (3-aminopropyl) triethoxysilane (APTES) were obtained from Shanghai Aladdin Industrial Co., Ltd. (Shanghai, China). 2-(N-morpholinyl) ethanesulfonic acid (MES) was purchased from Shanghai Sangong Bioengineering Technology Co., Ltd. (Shanghai, China). RPMI-1640 cell culture medium and fetal bovine serum (FBS) were purchased from Gibco (Invitrogen, CA, USA). 3,3′,5,5′-tetramethylbenzidine (TMB), dimethyl sulfoxide (DMSO), H_2_O_2_, Triton X-100 solution, paraformaldehyde, Rhodamine isothiocyanate (RBITC), and Hoechst 33258 were purchased from Sigma-Aldrich (St. Louis, MO, USA). 3-(4,5-Dimethyl-2-thiazole)-2,5-diphenyltetrazolium bromide (MTT) was obtained from Beijing Soleibao Technology Co., Ltd. (Beijing, China). Other reagents and chemicals were obtained from local commercial suppliers unless otherwise stated.

The fresh blood was obtained from KM mice (SPF grade) (Zhengzhou Huixing Experimental Animal Center, SCXK (Yu) 2019-0002). The animal study protocol was approved by the Animal Care and Use Committee of the College of Biological Engineering, Henan University of Technology (Ethic Approval Code: Haut202110−5).

### 2.2. Preparation of ^RGD^SPIO@MSN NPs

#### 2.2.1. Preparation of SPIO NPs

The oleic acid-modified magnetic SPIO NPs, i.e., O-γ-Fe_2_O_3_ NPs, served as the core in this present study, and were prepared by partially reduced co-precipitation in accordance with previously published methods (see Appendix A) [38].

#### 2.2.2. Preparation of SPIO@MSN NPs

The SPIO@MSN NPs were synthesized utilizing the surfactant-templated seeded growth method [39]. Detailed information on the preparation of SPIO NPs and SPIO@MSN NPs is provided in the Appendix A.

#### 2.2.3. Amination of SPIO@MSN NPs by APTES

The SPIO@MSN NPs were modified with (3-aminopropyl) triethoxysilane (APTES) to prepare NH_2_-SPIO@MSN NPs by a previously published method [40]. 50 mg of MSN was dispersed in 30 mL of ethanol, then 500 μL of APTES was added and the mixture was refluxed at 78 °C for 24 h. The obtained NH_2_-SPIO@MSN NPs were then harvested using a magnetic field, washed three times using ethanol, and dried at 60 °C overnight, in a vacuum.

#### 2.2.4. NH_2_-SPIO@MSN NPs and Cyclic RGD Peptides Were Activated by the Bifunctional Cross-Linker Sulfo-SMCC and Traut’s Reagent, Respectively

Cyclic RGD peptide-conjugated nanocarriers were prepared to employ sulfo-SMCC and Traut’s Reagent as crosslinking reagents. Briefly, 5 mL of cyclic RGD peptide solution, at a concentration of 1 mg/mL in borate buffer (pH 8, containing 2 mM EDTA), was mixed with 30 μL of Traut’s reagent (2 mg/mL), and the reaction was allowed to proceed for 60 min at room temperature (RT), to prepare cyclic RGD peptide-SH. At the same time, 250 μL of Sulfo-SMCC solution (5 mg/mL, dissolved in PBS) was added to a 5 mL NH_2_-SPIO@MSN suspension (10 mg/mL), and the reaction was allowed to proceed for 30 min at RT, to obtain sulfo-SMCC-SPIO@MSN.

#### 2.2.5. Cyclic RGD Peptide-Modified SPIO@MSN NPs

The cyclic RGD peptide-SH was mixed with the sulfate-SMCC-SPIO@MSN solution, and the reaction was allowed to proceed at RT for 60 min, to synthesize ^RGD^SPIO@MSN NPs. Finally, the obtained ^RGD^SPIO@MSN NPs were washed three times with PBS (pH 7.4, 0.1 M), resuspended in PBS (pH 7.4, 0.1 M), and stored at 4 °C for future use.

### 2.3. Stability Analysis of SPIO@MSN NPs and ^RGD^SPIO@MSN NPs in Different Media

The stability of fabricated SPIO@MSN NPs and ^RGD^SPIO@MSN NPs in different media was evaluated as follows: SPIO@MSN NPs and ^RGD^SPIO@MSN NPs, at a concentration of 1 mg/mL, were each dispersed in DI water, RPMI-1640 culture medium (Gibco, Invitrogen, CA, USA), and PBS solution (pH 7.4, 0.1 M). After ultrasonic dispersion, 300 μL of each nanoparticle dispersion was added to a cuvette. The stability was detected by evaluating the optical absorbance at 450 nm at predetermined intervals with a spectrophotometer (UV-3100PC).

### 2.4. Hemolytic Analysis of ^RGD^SPIO@MSN NPs

Fresh blood (3 mL) was collected using heparin as an anticoagulant and centrifuged for 5 min at 2500 rpm. After the supernatant was discarded, the obtained red blood cells (RBCs) were washed with saline solution (0.9%) and dispersed in the saline solution. Subsequently, 1 mL of the RBC suspension was added to ^RGD^SPIO@MSN NP solutions of various concentrations (25~200 μg/mL). After 4 h of stabilization, the products were centrifuged for 5 min at 2500 rpm, and then the absorbance at 540 nm was estimated for 200 μL of each supernatant to calculate the hemolysis ratio, with DI water as a positive control and normal saline (0.9%) as a negative control. The hemolysis rate was defined as in the following equation:(1)$$\mathrm{Hemolysis}\;\mathrm{rate}\left(\%\right)=\frac{OD_{c}-OD_{b}}{OD_{a}-OD_{b}}\times 100\%,$$
where *OD_a_* represents the absorbance of the positive control at 540 nm, *OD_b_* represents the absorbance of the negative control at 540 nm, and *OD_c_* represents the absorbance of the experimental samples at 540 nm.

### 2.5. Drug Loading and Drug Loading Content Analysis

The amino-functionalized SPIO@MSN NPs were resuspended in PBS solution (0.01 M, pH 7.4), followed by the addition of DOX at a mass ratio of 1:1. The mixture was then shaken at RT on a shaker for 24 h to form SPIO@MSN@DOX NPs, which were obtained via magnetic precipitation and washed three times with sterile PBS. Next, the conjugation of cyclic RGD peptides to SPIO@MSN@DOX NPs was performed according to the method described in Section 2.7. The absorbance of the supernatant was measured at 480 nm with a UV-Vis spectrophotometer (UV-3100PC). The drug loading content of DOX was calculated according to the following equation:(2)$$\mathrm{Loading}\;\mathrm{content}\;\left(\%\right)=\frac{W_{t}}{W_{s}}\times 100\%,$$
where *W_t_* represents the weight of DOX in the NPs, and *W_s_* represents the weight of the NPs.

### 2.6. Cell Culture

The human hepatocellular carcinoma HepG2 cell line, obtained from the Shanghai Cell Bank of the Chinese Academy of Sciences (Shanghai, China), was cultured in RPMI-1640 medium containing 10% FBS, 1% penicillin (100 U/mL), and 1% streptomycin (100 μg/mL) in a cell incubator at 37 °C with 5% CO_2_. The HepG2 cells were regularly subcultured every 3 d at a ratio of 1:3, to maintain cells in the exponential growth phase.

### 2.7. RBITC-Labeled SPIO@MSN NPs and ^RGD^SPIO@MSN NPs

SPIO@MSN NPs and ^RGD^SPIO@MSN NPs were labeled with rhodamine B-5-isothiocyanate (RBITC) according to the protocol reported previously [41]. The fluorescently labeled RBITC-SPIO@MSN NPs and RBITC-^RGD^SPIO@MSN NPs were observed under a fluorescence microscope. In addition, free RBITC, RBITC-SPIO@MSN NPs, RBITC-^RGD^SPIO@MSN NPs, unlabeled SPIO@MSN NPs and ^RGD^SPIO@MSN NPs were each diluted in DI water to scan the absorption spectra over the full 300~700 nm wavelength range, using an Ultra-micro spectrophotometer (NanoDrop 2000C).

### 2.8. Cellular Uptake Analysis

Cellular uptake of the RBITC-SPIO@MSN NPs and RBITC-^RGD^SPIO@MSN NPs was evaluated utilizing the fluorescence of the RBITC label. Briefly, HepG2 cells were seeded in a 24-microwell plate (1 × 10^5^ cells/well) and cultured for 48 h, and then 500 μL of RBITC-SPIO@MSN NPs or RBITC-^RGD^SPIO@MSN NPs (40 μg/mL) were added to individual wells. Residual nanoparticles were removed after a 4-h incubation in the dark, and the treated cells were washed three times using PBS (pH 7.4, 0.01 M). Then, the nuclei were stained for 20 min using Hoechst 33258, washed three times with PBS, and photographed with an inverted fluorescence microscope (Axio Observer 3). In addition, the treated HepG2 cells were collected and scanned by flow cytometry (FACS Calibur, BD, USA).

### 2.9. Cytotoxicity Analysis

The cytotoxicity of ^RGD^SPIO@MSN NPs was determined by the standard MTT assay [42]. Briefly, HepG2 cells were seeded in 96-microwell plates (3000 cells/well). After 24 h of culture, 200 μL of ^RGD^SPIO@MSN NPs and ^RGD^SPIO@MSN@DOX NPs, at different concentrations (0~200 μg/mL), were added to each well and incubated for another 24 h. The treated cells were washed with PBS (pH 7.4, 0.01 M), and then 200 μL of MTT solution (500 μg/mL) was added and incubated for 4 h in the dark. After the medium was removed, 150 μL of DMSO was added to each well and incubated for another 10 min. The absorbance of the solutions was measured at 490 nm using a microplate spectrophotometer (Tecan Spark).

### 2.10. Reactive Oxygen Species Assay

HepG2 cells treated with ^RGD^SPIO@MSN NPs and ^RGD^SPIO@MSN@DOX NPs were washed three times with PBS (pH 7.4, 0.01 M), then 1 mL of the fluorescent probe 2′, 7′-diacetyldichlorofluorofluorescein (DCFH-DA, 10 μM) was added and incubated for 20 min at RT in the dark. The treated HepG2 cells were then washed three times with a serum-free medium to remove the free DCFH-DA probe. The treated HepG2 cells were then digested with 0.25% EDTA-free trypsin, gently blown to make a cell suspension, centrifuged at 1000 rpm for 5 min, collected, and then washed three times with PBS. The levels of reactive oxygen species (ROS) in these treated cells were analyzed using flow cytometry (FACS Calibur, BD, USA).

### 2.11. Apoptosis Analysis of the Treated HepG2 Cells

To further evaluate the pro-apoptotic effect of various nanocomposites, treated HepG2 cells were stained using the bisbenzimide dye, Hoechst 33258, to visualize apoptotic cells. HepG2 cells were treated for 24 h with various nanocomposites, then the treated cells were fixed using 4% paraformaldehyde for 15 min, washed three times with PBS (pH 7.4, 0.01 M), and then stained with 200 μL of Hoechst 33258 (2.5 μg/mL) for 15 min in the dark at RT. The stained HepG2 cells were washed three times with PBS (pH 7.4, 0.01 M), and then photographed with an inverted fluorescence microscope (Axio Observer 3).

The apoptotic effect of various formulations on HepG2 cells was also determined with the Annexin V-FITC Apoptosis Detection Kit (Keygen Biotech, Nanjing, China). The treated HepG2 cells were digested with 0.25% EDTA-free trypsin and centrifuged at 1000 rpm for 5 min. The supernatant was discarded, the treated HepG2 cells were resuspended in 500 μL of PBS (pH 7.4, 0.01 M), and then 5 μL each of Annexin V-FITC and PI were added. After the mixture was incubated for 10 min at RT in the dark, the samples were analyzed by flow cytometry (FACS Calibur, BD, USA).

## 3. Results and Discussion

### 3.1. Synthesis and Characterization of ^RGD^SPIO@MSN@DOX NPs

Cyclic RGD peptide-modified magnetic mesoporous nanocarriers were prepared as illustrated in Figure 2. The procedures included (i) preparation of SPIO NPs, i.e., γ-Fe_2_O_3_ NPs as the magnetic core, using partial reduction co-precipitation; (ii) decoration with a mesoporous silicon layer, for the magnetic mesoporous silicon encapsulation of the γ-Fe_2_O_3_ NPs with large-aperture mesoporous silicon shells, using the improved Stöber method (SPIO@MSN NPs); (iii) amination of the magnetic mesoporous silicon by APTES (NH_2_-SPIO@MSN NPs); (iv) coupling of NH_2_-SPIO@MSN NPs with the bifunctional crosslinker sulfo-SMCC, and thiolation of cyclic RGD peptides with Traut’s Reagent; and (v) formation of cyclic RGD-targeted magnetic mesoporous nanocarriers by conjugating the sulfo-SMCC-SPIO@MSN NPs and RGD-SH peptides.

The fabricated nanocomposites were then characterized. Appendix A shows the morphology of the synthesized γ-Fe_2_O_3_ NPs, and the particle sizes were approximately 5~10 nm. As shown in Appendix A, the electron diffraction pattern corresponds to bright-field images, indicating that these particles were γ-Fe_2_O_3_ NPs. In addition, the SPIO@MSN NPs exhibited significant grooves and mesoporous channels, and the particle sizes were approximately 50~60 nm, with a uniform distribution (Figure 1B). The average hydrate particle sizes of various nanoparticles were 180.1, 280.6, 285.3, and 334.8 nm, respectively (Figure 1C). As shown in Figure 1D, the zeta potentials of SPIO NPs and SPIO@MSN NPs were approximately −6.79 mV and −13.4 mV, respectively, and the surfaces were negatively charged. The potential of NH_2_-SPIO@MSN NPs was about +7.8 mV, while that of ^RGD^SPIO@MSN NPs decreased to 2.46 mV after modification with RGD, which may be attributed to the fact that the cyclic RGD peptide is negatively charged in DI water. These data indicated that the targeted nanocarriers were successfully prepared.

The structure of the cyclic RGD peptide-targeted SPIO@MSN NPs was further characterized by FT-IR spectroscopy (Figure 2A). The results suggested that the ^RGD^SPIO@MSN NPs peaks at approximately 630 cm^−1^ corresponded to Fe-O vibrational bands. The absorption peaks at 3435 and 1625 cm^−1^ indicated O-H stretching, and deforming vibrations of adsorbed water, respectively. The peaks at 1081 cm^−1^, 460 cm^−1^ and 930 cm^−1^ could be attributed to Si-O-Si (a stretching vibration peak), Si-O-Si (a bending vibration peak) and Si-OH (a bending vibration peak), respectively. It is clear that the FT-IR spectrum of NH_2_-SPIO@MSN NPs contains signals associated with the N-H (1560 cm^−1^); however, a decrease in the characteristic peak of N-H bending was observed in the FT-IR spectrum of ^RGD^SPIO@MSN NPs, indicating that SPIO@MSN NPs and RGD peptides are conjugated via amine groups (Figure 2A). The XRD pattern of SPIO NPs (γ-Fe_2_O_3_) showed characteristic diffraction peaks at 2*θ* angles of 30.2°, 35.5°, 43.2°, 53.8°, 57.1°, and 62.7°, corresponding to γ-Fe_2_O_3_ (220), (311), (400), (422), (511) and (440) planes (Maghemite-Q, Card No. 25-1402), respectively [43]. In addition, SPIO@MSN NPs exerted a weak characteristic diffraction peak at the exact position of SPIO NPs due to the mesoporous silicon coating (Figure 2B). These results confirmed that SPIO@MSN NPs with a core-shell structure were successfully fabricated.

The pore diameter and specific surface area (SSA) of SPIO@MSN NPs were detected with N_2_ adsorption–desorption techniques, by the Barrett–Joyner–Halenda (BJH) and Brunauer–Emmett–Teller (BET) methods. The SSA of SPIO@MSN NPs was 1111.89 m^2^/g and the pore diameter was 4.63 nm, which meets the standard of mesoporous nanomaterials (Appendix A).

### 3.2. Analysis of Stability and Drug Loading Performance

The stability of prepared SPIO@MSN NPs and ^RGD^SPIO@MSN NPs was evaluated in three different media, DI water, PBS, and RPMI-1640 medium. As shown in Figure 2C, the stability of these two nanoparticles in the three media was pretty stable over 48 h. The biocompatibility of ^RGD^SPIO@MSN NPs was also tested. As shown in Figure 2D, the hemolysis rate of ^RGD^SPIO@MSN NPs was 3.89% at a concentration of 200 μg/mL, which is lower than 5%, indicating that the nanocarrier had good biocompatibility. Meanwhile, the DOX loading content was determined to be approximately 6.89% according to the standard curve of DOX and the calculation formula (Appendix A). Herein, DOX was loaded onto the nanocarrier SPIO@MSN NPs before modification by cyclic RGD peptide, due to the fact that the steric hindrance of RGD can block the surface of mesoporous pores and reduce the efficacy of drug loading.

**Figure 2 pharmaceutics-15-00276-f002:**
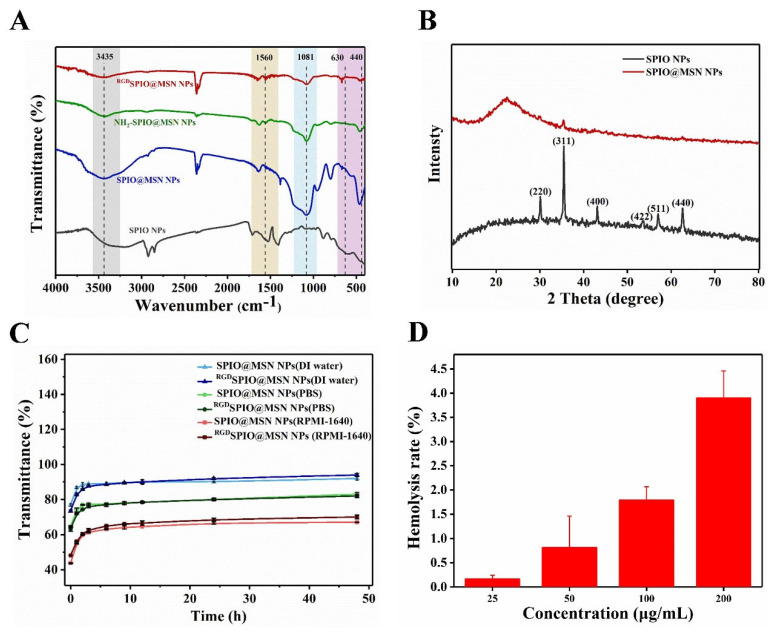
(**A**) FT-IR spectra of SPIO NPs, MSN NPs, SPIO@MSN NPs and ^RGD^SPIO@MSN NPs. (**B**) XRD patterns of SPIO NPs and SPIO@MSN NPs. (**C**) Stability assay of SPIO@MSN NPs and ^RGD^SPIO@MSN NPs in three different media, DI water, PBS and RPMI-1640 medium. (**D**) Assay of hemolysis rate of ^RGD^SPIO@MSN NPs.

### 3.3. RBITC-Labeled SPIO@MSN NPs and ^RGD^SPIO@MSN NPs

We used a fluorescent dye, the rhodamine derivative RBITC, to label SPIO@MSN NPs and ^RGD^SPIO@MSN NPs for further use in uptake analysis. SPIO@MSN NPs and ^RGD^SPIO@MSN NPs emitted strong red fluorescence (Figure 3A–C). In addition, the full-wavelength scanning results showed that the absorption peaks of RBITC-SPIO@MSN NPs and ^RGD^SPIO@MSN NPs appeared between 500 nm and 600 nm (Appendix A).

### 3.4. Cellular Uptake Evaluation

Cellular uptake was assayed by incubating HepG2 cells with RBITC-labeled SPIO@MSN NPs and ^RGD^SPIO@MSN NPs for 4 h. It can be seen from the fluorescence images that the red fluorescence intensity of targeted RBITC-^RGD^SPIO@MSN NP-treated cells was much stronger compared with that of non-targeted RBITC-SPIO@MSN NP-treated cells, indicating that modification with the cyclic RGD peptide could remarkably improve the ability of the SPIO@MSN NPs to target and enter HepG2 cells (Figure 4A,B).

Additionally, uptake of the SPIO@MSN NPs and ^RGD^SPIO@MSN NPs by HepG2 cells was further analyzed by flow cytometry. The uptake of NPs by cells in the ^RGD^SPIO@MSN NP group was higher than that in the SPIO@MSN NP group, further confirming the good targeting ability of RGD (Figure 4C,D). All these results suggest that the nanocarriers can successfully enter HepG2 cells and that modification with cyclic RGD peptides can significantly improve the targeting ability of the nanocarriers.

**Figure 4 pharmaceutics-15-00276-f004:**
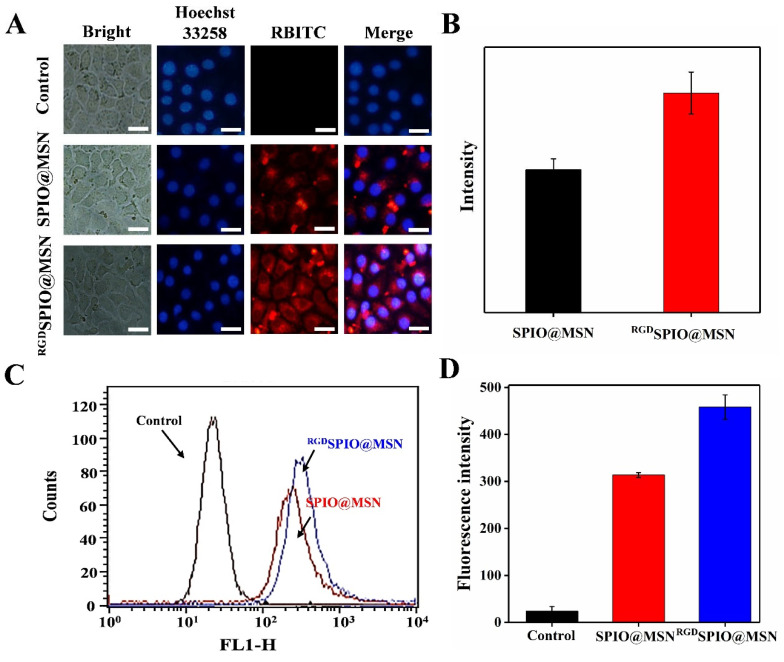
(**A**) Fluorescence images of HepG2 cells treated with RBITC-labeled SPIO@MSN NPs and ^RGD^SPIO@MSN NPs for 24 h, with cell nuclei stained with Hoechst 33258, and (**B**) the corresponding quantitative analysis, scalebar = 20 μm. (**C**) Assay of cellular uptake after 4 h treatment with various RBITC-labeled nanocarriers by flow cytometry, and (**D**) the corresponding quantitative analysis.

### 3.5. Cytotoxicity and Anti-Proliferative Activity Analysis

The MTT assay was performed to evaluate the survival rate of HepG2 cells treated with ^RGD^SPIO@MSN NPs, to investigate the cytotoxicity of this nanocomposite. The results suggested no apparent changes in cell viability with increasing concentrations of ^RGD^SPIO@MSN NPs, indicating that ^RGD^SPIO@MSN NPs have high biosafety (Figure 5A). In contrast, the proliferative activity of HepG2 cells treated with ^RGD^SPIO@MSN@DOX NPs significantly decreased with increasing concentrations of NPs. Remarkably, the cell survival rate was 40.29% when the ^RGD^SPIO@MSN@DOX NP concentration was 25 μg/mL, though the DOX concentration was only 1.72 µg/mL. In contrast, when the free DOX concentration reached 40 μg/mL, the cell survival rate was still as high as 62.48% (Appendix A). These results indicate that ^RGD^SPIO@MSN could achieve targeted delivery of the antitumor drug DOX to HepG2 cells, and enrichment of drug accumulation at the tumor site, thereby further inducing the cell survival rate to decrease (Figure 5B).

### 3.6. The Effect on the Level of Reactive Oxygen Species

ROS, a product of aerobic metabolism in vivo, is involved in lipid peroxidation, DNA strand breakage, protein modification and denaturation, and can affect intracellular signal transduction and gene expression. Many studies have suggested that the level of intracellular ROS is closely related to apoptosis [44,45,46]. In this report, we used the fluorescent probe DCFH-DA to label the cells treated with SPIO@MSN NPs and ^RGD^SPIO@MSN NPs, to explore the effect of these NPs on ROS levels, using flow cytometry. The ROS levels in cells treated with SPIO@MSN NPs, ^RGD^SPIO@MSN NPs and ^RGD^SPIO@MSN@DOX NPs for 24 h, were greater compared with the control group (Figure 5C,D). Among the groups, the fluorescence intensity of the ^RGD^SPIO@MSN NP group was higher than that of the SPIO@MSN NP group, indicating that the targeting effect of the cyclic RGD peptide was beneficial to the accumulation of ^RGD^SPIO@MSN NPs at the tumor site. Additionally, the ROS level was highest in the ^RGD^SPIO@MSN@DOX NP group.

**Figure 5 pharmaceutics-15-00276-f005:**
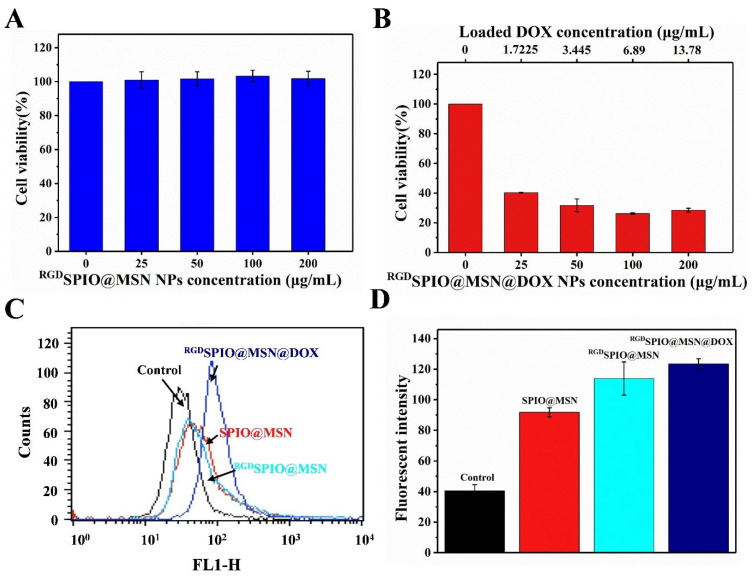
(**A**) Assay of cell viability after treatment for 24 h with ^RGD^SPIO@MSN NPs and (**B**) ^RG^SPIO@MSN@DOX NPs. (**C**) Analysis of the intracellular ROS levels after treatment with different nanoparticles for 24 h, by flow cytometry and (**D**) the corresponding quantitative analysis.

### 3.7. Analysis of Cell Apoptosis

Apoptosis is generally considered an essential marker of the efficacy of targeted therapies on cancer cells [47]. The main characteristics of morphological changes are cell shrinkage, chromosomal DNA fragmentation, nuclear fragmentation, and chromatin condensation [48]. To observe the nuclear fragmentation of apoptotic cells, the HepG2 cells were stained using the fluorescent dye Hoechst 33258, which can bind to the AT-rich DNA regions of chromatin [49], and thus the morphological changes of treated HepG2 cells can be observed with an inverted fluorescence microscope. In this study, HepG2 cells were treated with various fabricated nanocomposites for 24 h. As shown in Figure 6A, the HepG2 cells treated with the targeted ^RGD^SPIO@MSN@DOX NPs exhibited more typical changes, including chromatin condensation, perinuclear aggregation, nuclear fragmentation, and varying degrees of nuclear contraction. However, there was no obvious difference in nuclear changes between cells treated with SPIO@MSN NPs or ^RGD^SPIO@MSN NPs, and those in the control group. Quantitative analysis also clearly showed that the apoptosis rates of the groups treated with SPIO@MSN NPs and ^RGD^SPIO@MSN NPs were 7.19% and 11.07%, respectively; however, the apoptosis rate of the cells treated with ^RGD^SPIO@MSN@DOX NPs reached 33.53% (Figure 6B). These results indicated that the cyclic RGD peptide-targeted SPIO@MSN@DOX NPs we prepared could successfully deliver DOX into HepG2 cells and further induce apoptosis.

In addition, an Annexin V-FITC apoptosis kit was employed to evaluate the apoptotic effect on treated HepG2 cells. Annexin V is a sensitive indicator used to test the early apoptosis of cells [50]. Annexin V with FITC green fluorescence was used to label cells treated with various nanocomposites. The cell apoptosis rate was assessed using flow cytometry. The results show that with the SPIO@MSN NP and ^RGD^SPIO@MSN NP treatments, there was no obvious difference in the percentages of normal cells compared with untreated cells, indicating that these nanocarriers had no apparent toxicity to HepG2 cells. The apoptosis rate of the group treated with ^RGD^SPIO@MSN NPs was a little higher than that of the group treated with SPIO@MSN NPs, suggesting that the targeting of cyclic RGD peptide enhanced apoptosis. As seen from the results, treatment of HepG2 cells with ^RGD^SPIO@MSN@DOX NPs resulted in a low cell survival rate of 67.38%, indicating that the cyclic RGD peptide-modified vector could precisely target HepG2 cells for transport of the antitumor drug DOX and cause a large accumulation of the drug, promoting apoptosis in HepG2 cells (Figure 7).

## 4. Conclusions

We have successfully constructed a targeted delivery vector, cyclic RGD peptide-conjugated SPIO@MSN NPs, which was used for loading the antitumor drug DOX and treating hepatocellular carcinoma HepG2 cells. The ^RGD^SPIO@MSN NPs displayed high drug-loading efficiency, good stability, and biosafety. In vitro study demonstrated that the targeted ^RGD^SPIO@MSNs NPs could remarkably improve uptake by HepG2 cells, inhibit proliferation, and enhance cell apoptosis after being loaded with DOX. ROS analysis further confirmed that this nanosystem could increase ROS levels in HepG2 cells and promote cell apoptosis. Therefore, cyclic RGD peptide-conjugated SPIO@MSN@DOX NPs should be of great significance for the delivery of antitumor drugs and the treatment of hepatic carcinoma in the future.

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
