# Peer review of "Synergistic Pro-Apoptotic Effect of a Cyclic RGD Peptide-Conjugated Magnetic Mesoporous Therapeutic Nanosystem on Hepatocellular Carcinoma HepG2 Cells"

_pharmaceutics, 2023, doi:10.3390/pharmaceutics15010276_

Round 1

Reviewer 1 Report

This manuscript (pharmaceutics-2129308) by Zhao et al., described a magnetic mesoporous nanoparticles decorated with cyclic RGD as a doxorubicin delivery system toward HepG2 cells. Overall, I ask major revision to authors to be accepted in pharmaceutics due to the below reasons.

1. In this study, what are the advantages of the use of SPIO NPs? SPIO NPs-mesoporous silica vs. just mesoporous silica NPs.

2. In methods, please check the full name of APTES.

3. Please report the reaction yield of cyclic RGD in the conjugation to NPs.

4. Is there any reason why DOX is not loaded after RGD peptide is conjugated to the NPs?

5. There are no standard deviation in Figure 1D, Figure S3, Figure 2C,D, Figure 4D, and Figure 7B.

6. Does the DOX loading rate indicate DOX loading contents?

7. In Figure 5B, what dose the concentration in the x-axis indicate; DOX concentration or NP concentration? Both concentration should be indicated in the x-axis.

8. Please use free DOX and SPIO@MSN@DOX as control groups in Figure 5B.

9. It is difficult to see difference of chromatic condensation, perinuclear aggregation, and nuclear fragmentation among the samples in Figure 6A.

10. Y-axis of Figure 7B should be adjusted to allow readers to read the percentage of each graph.

11. Is the apoptotic rate in Figure 6B correlated with the sum of early apoptosis and late apoptosis in Figure 7B?

Reviewer 2 Report

In this work the authors presented the synthesis of mesoporous nanoparticles with SPIO core loaded with doxorubicin, for the treatment of HCC. The work is interesting but some points should be better clarify before the publication. My major concern it's about the lack of tests on normal cells.

-Is not clear the use of SPIONPs and their activity in this particular type of nanoparticles. Which is the specific effect of SPION on tumors in general? Which is the advantage in using SPIONPs as core?

-"RGD peptide (arginine-glycine-aspartic acid, Arg-Gly-Asp), highly expressed in tumor cells and blood vessels, can 56 specifically bind to αvβ3 integrin [30–34]." I found this sentence not fully correct, since is better to explain that RGD peptide is a common aminoacidic sequence found in many proteins overexpressed in tumor tissues.

-line 168: "Then, the treated HepG2 cells were digested using 0.25% EDTA-free 168 trypsin, collected, and washed three times using PBS." I suppose that cells were just detached from plates and not completely digested, Please adjust this sentence.

-For nanoparticles synthesis, I suggest to nominate the reaction steps in the paragraph in the same way as in the Scheme 2 

-Figure 2D: Hymolisis is Hemolysis

-Line 232: "... and that of RGDSPIO@MSN NPs was better." Better respect to what?

-Figure 3A: The cells reported in the figure are clearly dead. Please change the figure with appropriate cells images.

-"However, the normal cells treated with RGD SPIO@MSN@DOX NPs were as low as 67.38%, and the apoptotic proportion of cells significantly increased after RGDSPIO@MSN NPs were loaded with DOX." This sentence is unclear. Were are data about normal cells? Please clarify this point. The authors say that RGD SPIO@MSN@DOX NPs were as low as 67.38% but is not clear what do they refer to. 

-Figure 6: The authors should try to show the dox signal since dox is excitable and visible with a confocal microscope.

Round 2

Reviewer 1 Report

Comments:

The author sincerely responded to the reviewer’s concerns. I would like to ask authors to revise manuscript slightly before accepted.

1. In the standard curve (Figure S3), you can input standard deviation. Please check other numerous papers

2. Regarding point 4, the response is not reasonable. During reaction with RGD, the DOX in the mesoporous NPs can be diffused out to the media. Nevertheless, the authors' method (DOX loading and then RGD modification) is valuable because steric hindrance of RGD can block the surface of mesoporous pores in the method 2 (RGD modification and then DOX loading), which reduce the efficacy of drug loading. I would like to ask to add this kind of discussion into the manuscript.
